# MAGiC: Attributed Graph Generation via Mixed-type Diffusion and Coarsening

## Abstract

Many domains, like social and document networks, model relationships as graphs with rich node attributes and large node counts. However, modern graph generators cater to the discreteness of edge connectivity, at the expense of only allowing categorical node labels, and are limited to relatively small graphs, like molecules, due to scalability challenges. To overcome such challenges, we propose MAGiC, a framework that enables graph diffusion with mixed-type node attributes and improves scalability even in unattributed graph scenarios. At the core of MAGiC, a novel mixed-type diffusion joins discrete diffusion for the graph structure with continuous diffusion for node attribute embeddings in a single model. It enables the generation of nodes with rich attributes while maintaining the graph structure quality benefits of discrete diffusion. Alongside it, we propose an invertible coarsening algorithm and a structure-aware attribute encoder that boost scalability, reducing diffusion memory and computation costs. We evaluate MAGiC against baselines combining unattributed graph and tabular generation on three datasets with rich node attributes. Our solution is on average $12.9\times$ better at capturing attribute–structure interaction and $25.2\%$ better at downstream machine learning tasks. Concurrently, we maintain competitive synthesis quality for simple graphs with single categorical node labels. Moreover, MAGiC's coarsening (and attribute encoder) consistently reduces inference time by $2.5\times$ for simple and rich graphs.

## 1 Introduction

Graphs are essential in modeling relationships between entities, like social users (Rozemberczki & Sarkar, 2021), hyperlinked documents (Hu et al., 2020), or financial transactions (Altman, 2021), in many data-intensive applications. Alongside structural relationships, nodes and edges often store additional information as attributes, increasing the complexity and power of graph representations. Figure 1 shows an example graph with rich node attributes, where nodes have multiple categorical and continuous attributes. Crucially, the graph's node attributes and structural connectivity depend on each other. For instance, users from the same "Region" attribute are more likely to be connected.

State-of-the-art graph generators employ diffusion models for maximum synthesis quality (Cao et al., 2024). In images, pixels model inherently continuous color scales, making them a good fit for *continuous* diffusion models (Ho et al., 2020). However, graph structures are discrete (edges exist or do not) and sparse (most possible node pairs are not connected), requiring a different approach. *Discrete* diffusion models (Vignac et al., 2023; Chen et al., 2023; Qin et al., 2024), emerge as a better option to preserve key global graph structural properties. However, discrete noising limits the type of node attributes to categorical only. As such, they focus on use-cases like molecular-generation where nodes have strictly categorical, i.e., discrete, labels. Moreover, diffusion graph models are generally expensive to scale regarding computation and memory, as they work directly in the adjacency matrix space, scaling quadratically with node count. State-of-the-art attributed models (Vignac et al., 2023; Jo et al., 2024) limit their evaluation to graphs with under two hundred nodes. Existing work on reducing memory usage comes at the cost of extra processing time or quality drop, and only considers nodes with no attributes or discrete labels (Bergmeister et al., 2024; Kong et al., 2023).

We propose MAGiC, the first **M**ixed-type **A**ttributed **G**raph **D**iffusion model with **C**oarsening for generating graphs with rich node attributes at scale via lower computation and memory costs. To jointly model rich node attributes, represented as continuous embeddings, and the discrete graph

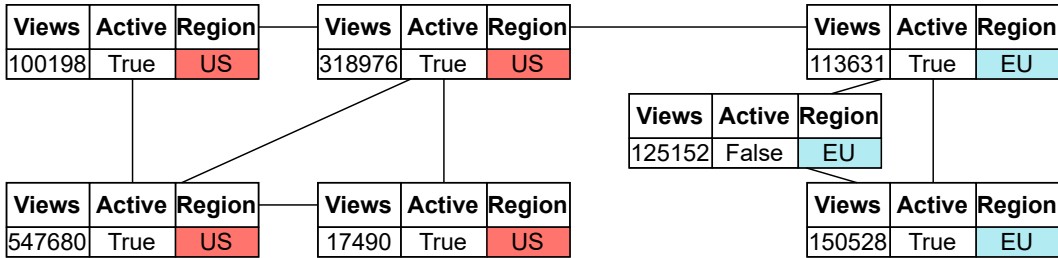

Figure 1: Social network example: multi-attributed nodes are users; edges are friendship relations. Users mainly connect with others from the same "Region" (shown as highlighted attribute).

structure, MAGiC harnesses *mixed-type diffusion*, combining continuous and discrete diffusion processes, for nodes and edges, respectively. To improve scalability, MAGiC consists of a novel *invertible coarsening* technique that compresses the graph structure into a smaller attributed graph with reduced memory footprint and lower computational cost, allowing lossless reconstruction. Finally, to efficiently handle high-dimensional node attributes, MAGiC encodes them into smaller latent-space embeddings via a *structure-aware variational autoencoder* (sVAE). To avoid potential negative biases related to the input order of nodes or edges, we prove the permutation invariance of all MAGiC's components.

We evaluate MAGiC against joint graph and tabular diffusion models on four groups of metrics targeting graph structure, attribute quality, inter-dependency between structure and attributes, and downstream task performance. MAGiC outperforms the baselines in all metric groups, capturing on average attribute-structure interaction $12.9\times$ better and improving the accuracy of downstream machine learning tasks by an average of $25.2\%$. It also reduces memory consumption and accelerates synthesis by $2.5\times$ compared to the baselines, even in unattributed settings.

In summary, our contributions are as follows:

- A novel mixed-type graph diffusion model that jointly optimizes the synthesis of discrete graph structure and continuous node attribute embeddings.
- An invertible coarsening scheme that, together with a structure-aware attribute encoder, reduces memory and computational costs of handling graph representations.
- Proofs for permutation symmetry properties of MAGiC, which ensure learning and synthesis are independent of node and edge ordering.
- Evaluation on three datasets with rich node attributes against existing graph diffusion baselines augmented with tabular attribute generators.

We provide our anonymized code at: `anonymous.4open.science/r/MAGiC-5615`.

## 2 BACKGROUND AND RELATED WORK

**Graph synthesizers** span various generative modeling techniques, including variational autoencoders (VAEs), generative adversarial networks (GANs), and autoregressive or diffusion models. Approaches leveraging VAEs, such as GraphVAE (Simonovsky & Komodakis, 2018), struggle to harness the latent space for graphs with more than a couple of dozen nodes. Earlier autoregressive models, like GraphRNN (You et al., 2018), iteratively synthesize arbitrarily large graphs, but their reliance on node order negatively affects output quality. GAN formulations, like SPECTRE (Martinkus et al., 2022), cannot match the quality of diffusion-backed models, and an inherently greater difficulty in training a discriminator/generator pair reduces their applicability. Graphs often exhibit hierarchical properties, which some models explicitly exploit within their modeling techniques. For instance, HiGen (Karami, 2024) and HGGT (Jang et al., 2024) are transformer-based models that decompose the adjacency matrix based on clusters of connected nodes or recursive splits. To maximize generation quality, we formulate MAGiC as a diffusion model.

**Diffusion models** Ho et al. (2020) have recently been at the forefront of high-quality synthetic data generation for many modalities, including graphs (Zhu et al., 2022). For graph diffusion models, the *noise type* is an important differentiator, which can be continuous (Jo et al., 2022), as for most

other modalities (Fuest et al., 2024), or discrete (Chen et al., 2023; Haefeli et al., 2022). The latter matches the discrete nature of graph connectivity, better modeling properties like sparsity. Shi et al. (2025), One of the few other mixed-type formulations, models numerical and categorical tabular data with continuous and discrete diffusion, respectively. MAGiC adapts mixed-type diffusion for generic graph generation.

**Attributed graphs** require incorporating node/edge attribute generation for synthetic graphs, increasing the problem complexity. Hence, such generators are limited to smaller graphs, even if they only integrate single-label nodes or edges. DiGress (Vignac et al., 2023) proposes a discrete denoising diffusion approach that predicts individual nodes and edges to generate graphs at scales up to two hundred nodes. Alternatives focused on scalability are also limited to categorical node labels. GraphMaker (Li et al., 2024) investigates discrete diffusion on larger graph structures. GraphARM (Kong et al., 2023) mixes discrete diffusion with autoregressive generation to improve sampling time, but the latter introduces an inherent bias related to node ordering. Jo et al. (2024) applies diffusion based on a mixture of bridge processes conditioned on the training samples. Its architecture allows synthesizing low-dimensionality discrete and continuous node attributes, but its overall generation quality is lower than recent discrete diffusion models (Qin et al., 2024). With MAGiC, we allow generating graphs with rich node attributes and improve scaling for larger graphs.

**Coarsening** is a technique for reducing graph dimensionality while preserving key properties. Many versions consist of fixed algorithms (Purohit et al., 2014), but newer works explore variants that are learnable through neural networks (Cai et al., 2021). All such methods operate on the graph structure, for example, striving to preserve similar spectral properties (Jin et al., 2020), and some additionally incorporate node attributes (Kumar et al., 2023). Coarsening is mainly used for analysis tasks on graphs, like prediction, meaning that only a unidirectional mapping is necessary. MAGiC is the first to employ coarsening to create an encoding that remains a valid graph with a corresponding lossless inverse coarsening.

## 3 MAGiC

At the heart of MAGiC is a *mixed-type diffusion* model, ① in Figure 2, that captures node attributes and structural connectivity in a unified manner. The *invertible coarsening* ② and *structure-aware attribute encoder* ③ act as preprocessing steps boosting the efficiency of diffusion. By simultaneously optimizing the denoising and embedding processes over both modalities, MAGiC accurately captures critical dependencies between node features and graph connectivity, essential for generating realistic attributed graphs. Formally, we tackle the generation of undirected graphs with rich node attributes $\mathbf{G}^r = (\mathbf{V}, \mathbf{M})$ where $\mathbf{V} \in \mathbb{R}^{n' \times r}$ are the $n'$ node attributes encoded as embeddings of size $r$, and $\mathbf{M} \in \{0, 1\}^{n' \times n'}$ is the adjacency matrix. For training, first, our *structure-aware attribute encoder* reduces the size of node attributes, transforming $\mathbf{G}^r$ into $\mathbf{G}^s = (\mathbf{Z}, \mathbf{M})$ with $\mathbf{Z} \in \mathbb{R}^{n' \times s}$, and $s \ll r$. Then, the *invertible coarsening* maps $\mathbf{G}^s$ to $\mathbf{G} = (\mathbf{X}, \mathbf{E})$ where $\mathbf{X} \in \mathbb{R}^{n \times f}$ are the attributes of $n < n'$ nodes represented via embeddings of size $f = 2r$, and $\mathbf{E} \in \mathcal{E}^{n \times n}$ is an adjacency matrix of edges of possible types $\mathcal{E}$ (including no edge). Finally, the *mixed-type diffusion* learns to synthesize compact graphs $\mathbf{G}$. During sampling, MAGiC generates $\mathbf{G}$, uncoarsens it, and decodes it back to the initial $\mathbf{G}^r$. Figure 2 shows the integration of MAGiC's three key components:

- Mixed-type Diffusion ① – a joint continuous-discrete denoising process over node embeddings and graph structure.
- Invertible Coarsening ② – a lossless bidirectional compression of the graph into a smaller one for efficient diffusion.
- Structure-Aware Attribute Encoder ③ – a variational encoder for node attributes that complement coarsening and harnesses node neighbor information.

These components make MAGiC suitable for generating graphs with richer node attribute sets compared to previous diffusion-based generators, while improving scalability. While we focus on integration with our mixed-type diffusion model, coarsening and sVAE are usable with any attributed graph generator. In Section 3.1 we first discuss the details of *mixed-type diffusion*, as the main component of our framework. We then describe the *invertible coarsening* in Section 3.2, followed by the *structure-aware attribute encoder* in Section 3.3. Finally, in Section 3.4, we cover theoretical results regarding symmetry properties of the different components, and their interaction. We

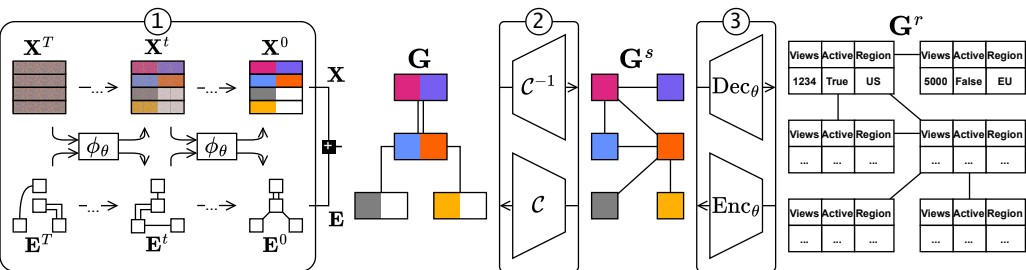

Figure 2: *MAGiC* overview: ① attribute encoder, ② coarsening, ③ mixed-type diffusion

summarize all notation from the manuscript in Appendix A, and include additional implementation notes in Appendix B.1.

### 3.1 MIXED-TYPE GRAPH DIFFUSION

We describe a mixed-type diffusion process that synthesizes graphs $G = (\mathbf{X}, \mathbf{E})$ with *nodes* $\mathbf{X} \in \mathbb{R}^{n \times f}$, and *edges* $\mathbf{E} \in \mathcal{E}^{n \times n}$. This formulation addresses a core modeling challenge: the discrete graph structure and continuous embeddings of rich node attributes each benefit from a noising approach tailored to their own modalities. Naively applying Gaussian noise to discrete edge connectivity compromises graph structure generation quality. Meanwhile, discrete diffusion is naturally unfit for continuous embeddings. Our solution unifies continuous diffusion for node embeddings with discrete diffusion for edge types. Optimizing both diffusion steps with a single model allows for capturing critical attribute-structure interactions and maintains a valid graph representation at each denoising step. Moreover, it makes training and generation more efficient due to the inherent parameters sharing, modeling common knowledge between nodes and edges.

Our formulation is a denoising diffusion probabilistic model DDPM (Ho et al., 2020). A forward process injects noise into each clean graph $\mathbf{G}^0 = (\mathbf{X}^0, \mathbf{E}^0)$ over $T$ consecutive time steps, producing increasingly corrupt versions $(\mathbf{G}^1, ..., \mathbf{G}^T)$ approaching a noise distribution. A noisy graph $\mathbf{G}^t = (\mathbf{X}^t, \mathbf{E}^t)$ has $\mathbf{X}^t \in \mathbb{R}^{n \times f}$ and $\mathbf{E}^t \in [0, 1]^{n \times n \times |\mathcal{E}|}$. $\mathbf{E}^t$ is an expanded noisy adjacency matrix, with entries $e_{ij}^t \in [0, 1]^{|\mathcal{E}|}$ encoding probability distributions over edge types between nodes $i$ and $j$. $\mathbf{G}^0$ is equivalent to $\mathbf{G}$, with $\mathbf{X}^0 = \mathbf{X}$ and $\mathbf{E}^0$ a one-hot encoding of $\mathbf{E}$. A neural network $\phi_\theta$ with parameters $\theta$ learns to approximate the reverse process of denoising $\mathbf{G}^t$ into $\mathbf{G}^0$.

**Forward Diffusion:** The forward noising process is a composition of two independent Markov chains for node embeddings $\mathbf{X}^t$ and edge types $\mathbf{E}^t$. For **node embeddings** $\mathbf{X}^t$, we apply variance-preserving Gaussian noise at each time step transition $t-1 \rightarrow t$ in a standard DDPM fashion. Thus, following Ho et al. (2020), for a noise schedule $\alpha_t \in (0, 1)$, $\bar{\alpha}_t = \prod_{i=1}^{t} \alpha_i$, and noise $\epsilon \sim \mathcal{N}(\mathbf{0}, \mathbf{I})$ sampled from a standard Gaussian distribution, we have a closed-form transition from $\mathbf{X}_0$ to $\mathbf{X}_t$:

$$\left( q_{\mathbf{X}}(\mathbf{X}^t \mid \mathbf{X}^0) = \mathcal{N}(\sqrt{\bar{\alpha}_t}\,\mathbf{X}^0, (1 - \bar{\alpha}_t)\,\mathbf{I}) \right) \leftrightarrow \left( \mathbf{X}^t = \sqrt{\bar{\alpha}_t}\,\mathbf{X}^0 + \sqrt{1 - \bar{\alpha}_t}\,\epsilon \right) \tag{1}$$

For **edges** $\mathbf{E}^t \in [0, 1]^{n \times n \times |\mathcal{E}|}$, we follow the discrete forward noising variant from Vignac et al. (2023). As with nodes, each edge $e_{ij}$ is noised independently. The matrix $\mathbf{Q}^t \in [0, 1]^{|\mathcal{E}| \times |\mathcal{E}|}$ encodes the transition distribution between edge types as a function of the target noise distribution $\mathbf{m}_E$ (the prior distribution of edge types in training data), and $\alpha_t$. Like node embeddings, we can sample any $\mathbf{E}^t$ from the initial data $\mathbf{E}^0$ in a closed form via $\bar{\mathbf{Q}}^t = \prod_{i=1}^{t} \mathbf{Q}^i$:

$$q_{\mathbf{E}}(\mathbf{E}^t \mid \mathbf{E}^0) = \prod_{i,j} \left( e_{ij}^0\,\bar{\mathbf{Q}}^t \right) \tag{2}$$

The **joint distribution** is thus:

$$q_{\mathbf{G}}(\mathbf{G}^t \mid \mathbf{G}^0) = q_{\mathbf{X}}(\mathbf{X}^t \mid \mathbf{X}^0)\,q_{\mathbf{E}}(\mathbf{E}^t \mid \mathbf{E}^0)$$

We follow the consensus from existing diffusion works (Yang et al., 2024) where the forward process should be as localized as possible and converge to a simple to model target noise distribution even

**Algorithm 1** Mixed-type Diffusion Train Step

    **Input**: model $\phi_\theta$, graph $\mathbf{G} = (\mathbf{X}, \mathbf{E})$
1: $t \sim \mathcal{U}(1, \ldots, T)$; $\epsilon \sim \mathcal{N}(\mathbf{0}, \mathbf{I}_n)$
2: $\mathbf{X}^t = \sqrt{\bar{\alpha}_t}\, \mathbf{X} + \sqrt{1 - \bar{\alpha}_t}\, \epsilon$   {Equation (1)}
3: $\mathbf{E}^t_{ij} \sim \prod_{i,j} \left( e_{ij}\, \bar{\mathbf{Q}}^t \right)$     {Equation (2)}
4: $\hat{\epsilon}, \hat{p}_E \leftarrow \phi_\theta(\mathbf{X}^t, \mathbf{E}^t)$
5: $\mathcal{L}_{cont} \leftarrow \|\hat{\epsilon} - \epsilon\|^2_2$     {Equation (5)}
6: $\mathcal{L}_{disc} \leftarrow \sum_{i,j} \mathrm{CE}(\hat{\mathbf{p}}_{i,j}, \mathbf{E}_{i,j})$ {Equation (6)}
7: Optimize $\theta$ via $\mathcal{L}_{\text{cont}} + \lambda\, \mathcal{L}_{\text{disc}}$

**Algorithm 2** Mixed-type Diffusion Sampling

    **Input**: model $\phi_\theta$
1: $\mathbf{X}^T \sim \mathcal{N}(\mathbf{0}, \mathbf{I})$; $\mathbf{E}^T_{ij} \sim \prod_{i,j} \mathbf{m}_E$
2: **for** $t \leftarrow T, \ldots, 1$ **do**
3:     $\hat{\epsilon}, \hat{p}_E \leftarrow \phi_\theta(\mathbf{X}^t, \mathbf{E}^t)$
4:     $\mathbf{z} \sim \mathcal{N}(\mathbf{0}, \mathbf{I})$
5:     $\mathbf{X}^{t-1} \leftarrow$ Equation (3)
6:     $\mathbf{E}^{t-1} \sim$ Equation (4)
7: **end for**
8: **Return** $(\mathbf{X}^0, \mathbf{E}^0)$

when working with homogeneous data structures. For instance, noise in image generation is applied independently per pixel, and the noise distribution is a Gaussian. Doing so helps to keep training efficient, as noise is fast, and ensures the model starts denoising from an easy-to-model distribution. In our case, keeping the two noise processes independent retains such properties.

**Reverse Diffusion and Training Objective:** Given a noisy node representation at step $t$ and $\epsilon$ dictating the added noise, the representation at $t$–1 is (Ho et al., 2020):

$$\mathbf{X}^{t-1} = \frac{\mathbf{X}^t}{\sqrt{\alpha_t}} - \frac{(1 - \alpha_t)\, \epsilon}{\sqrt{\alpha_t\, (1 - \bar{\alpha}_t)}} + \sqrt{\frac{(1 - \bar{\alpha}_{t-1})\, (1 - \alpha_t)}{1 - \bar{\alpha}_t}}\, \mathbf{z} \tag{3}$$

where $\mathbf{z} \sim \mathcal{N}(\mathbf{0}, \mathbf{I})$ for $t > 1$ else 0 dictates the partial noise added back to the representation. For edges, taking $\mathbf{Q}^{t'}$ as the transpose of $\mathbf{Q}^t$ and edge type $e \in \mathcal{E}$, we have (Vignac et al., 2023):

$$q_{\mathbf{E}}(e^{t-1}_{ij} \mid e^t_{ij},\, e_{ij} = e) \propto e^t_{ij} \mathbf{Q}^{t'} \odot e^0_{ij} \bar{\mathbf{Q}}^{t-1}$$

When considering all possible edges and their types $\mathcal{E}$, we obtain a proper probability distribution:

$$q_{\mathbf{E}}(\mathbf{E}^{t-1} \mid \mathbf{E}^t) = \prod_{i,j} \sum_{e \in \mathcal{E}} q(e^{t-1}_{ij} \mid e^t_{ij},\, e_{ij} = e)\, p_{ij}(e) \tag{4}$$

The denoising network $\phi_\theta$ approximates the joint reverse process $p_\theta(\mathbf{G}^{t-1} \mid \mathbf{G}^t) = p_\theta(\mathbf{X}^{t-1} \mid \mathbf{G}^t)\, p_\theta(\mathbf{E}^{t-1} \mid \mathbf{G}^t)$. Since $\phi_\theta$ accounts for dependencies between nodes and edges, it entangles the approximated reverse process over the two components. For node embeddings, we optimize:

$$\mathcal{L}_{\text{cont}} = \mathbb{E}_{t, \mathbf{X}^0, \epsilon} \left[ \|\hat{\epsilon} - \epsilon\|^2_2 \right] \tag{5}$$

For edge types, we formulate the problem as multi-class classification with cross-entropy (CE) loss:

$$\mathcal{L}_{\text{disc}} = \mathbb{E}_{t, \mathbf{E}^0} \left[ \sum_{i,j} \mathrm{CE}(\hat{\mathbf{p}}_{i,j}, \mathbf{E}_{i,j}) \right] \tag{6}$$

The mixed diffusion training objective combines both losses $\mathcal{L}_{\text{mixed}} = \mathcal{L}_{\text{cont}} + \lambda\, \mathcal{L}_{\text{disc}}$ where $\lambda > 0$ balances the relative importance of the continuous and discrete components. The joint objective and the shared network architecture capture crucial attribute-structure interactions.

**Modelling Setup:** Algorithm 1 describes a training step in the model. We noise the input graph $\mathbf{G}$ up to an arbitrary time step $t$. Note that $\mathbf{E}^t$ should be symmetrized such that $\mathbf{E}^t_{ij} = \mathbf{E}^t_{ji}$. Then, given the noisy graph, the model predicts the noise added to nodes and the probability distribution for each edge's type. Finally, the model's parameters get optimized in accordance with $\mathcal{L}_{mixed}$. Algorithm 2 contains sampling details. We first sample fully noised node attributes $\mathbf{G}$ from the Gaussian prior and a one-hot encoded discrete adjacency matrix from the prior of edge types within training data. Then, over $T$ steps, we use the trained model's prediction to partially denoise both components until reaching a new clean synthetic graph from the training distribution.

## 3.2 INVERTIBLE COARSENING

The computational complexity and memory footprint of diffusion models scale quadratically with the number of nodes, limiting their applicability to large-scale graph generation. Simultaneously,

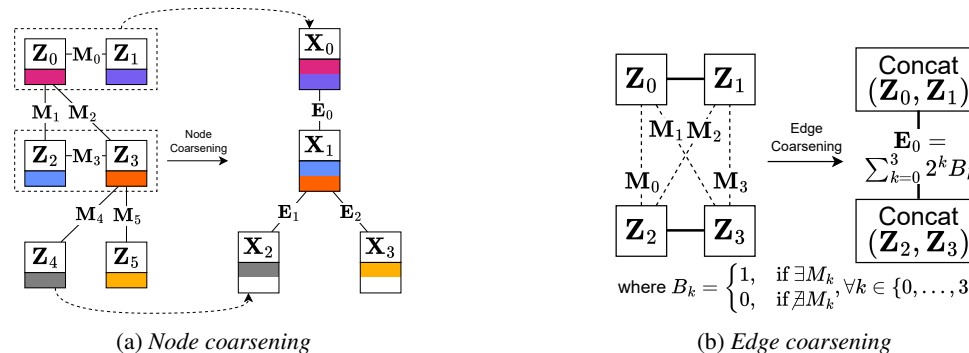

(a) *Node coarsening*             (b) *Edge coarsening*

Figure 3: Coarsening overview

node attributes' computational complexity and memory footprint scale linearly with their size. We cannot apply current graph coarsening techniques to graph diffusion, as their design is limited to irreversible (unidirectional) mapping, preventing reconstruction of the larger initial graph. We improve the scalability of graph diffusion with an **invertible coarsening mechanism** that shrinks the input graph by merging adjacent nodes and storing reconstruction information as expanded node attributes and additional node types. Reducing the number of nodes at the cost of more features enables further compression given by the sVAE (see section 3.3), creating even more space for computation and memory costs improvements.

Given a graph $\mathbf{G}^s = (\mathbf{Z}, \mathbf{E}^s)$ with nodes $\mathbf{Z} \in \mathbb{R}^{n' \times s}$ and binary adjacency matrix $\mathbf{E}^s \in \{0, 1\}^{n' \times n'}$, the coarsening function $\mathcal{C} : \mathbf{G}^s \mapsto \mathbf{G}$ produces a compressed graph $\mathbf{G} = (\mathbf{X}, \mathbf{E})$ with merged nodes $X \in \mathbb{R}^{n \times f}$ and attributed edges $E \in \mathbb{N}^{n \times n}$, where $n \approx \frac{n'}{2}$, and $f = 2s$. The inverse function $\mathcal{C}^{-1} : \mathbf{G} \mapsto \mathbf{G}^s$ maps $\mathbf{G}$ to $\mathbf{G}^s$, up to node permutation. Appendix B.2 describes the coarsening and inverse coarsening steps more formally, along with pseudocode.

**Node Merging:** The coarsening procedure operates through a greedy node merging algorithm based on the embedding similarity of neighbor nodes. Figure 3a visualizes the process. We define a distance metric $\mathcal{D}(z_i, z_j) = \|z_i - z_j\|_1$ over pairs of node embeddings $z_i$ and $z_j$ from $\mathbf{G}^s$. We merge neighboring node pairs $(z_i, z_j)$ in ascending order of $\mathcal{D}$, with each merged node represented as:

$$x_k = \text{Concat}(z_i, z_j) \in \mathbb{R}^f \tag{7}$$

We zero-pad embeddings of unpaired nodes to maintain consistent dimensionality. We show in Appendix C, as part of Proposition 4, that coarsening is permutation invariant.

**Edge Encoding**: The critical challenge for our invertible coarsening is preserving the original connectivity information within the compressed representation. Since we merge nodes in pairs, each coarsened edge $e_{ij}$ from $\mathbf{E}$ models the connections among two pairs of original nodes (four nodes total). Within a pair, the edge between its nodes is implicit. Between two pairs, we encode connectivity as the binary encoding of the four potential edges in a categorical edge attribute $e_{ij} \in \mathbb{N}^{15}$:

$$e_{ij} = \sum_{k=0}^{3} b_k \cdot 2^k \tag{8}$$

where $b_k \in \{0, 1\}$ indicates the presence of the $k$-th edge between nodes (see Figure 3b example).

**Inversion:** The inverse coarsening $\mathcal{C}^{-1}$ operates by decomposing each merged node $x_k$ into its constituent embeddings $(z_i, z_j)$ and decoding each edge attribute $e_{ij}$ to recover the original connectivity pattern. Specifically, we know each merged node splits into two connected nodes. We reverse the binary encoding between pairs of merged nodes to find the connectivity between initial nodes from different pairs. As such, in Appendix C we prove that:

**Proposition 1.** *MAGiC's coarsening is invertible, up to node permutation.*

Specifically, for graphs with the *unique distances* property (Definition 1), inverse coarsening is injective: each permutation $\pi$ over a coarsened $\mathbf{G}$ maps to a unique uncoarsened $\mathbf{G}_s$. For an uncoarsened graph $\mathbf{G}^u$ without the property, $\mathcal{C}^{-1}(\mathcal{C}(\mathbf{G}^s))$ will still be a permutation $\pi$ of $\mathbf{G}^s$, albeit not unique.

### 3.3 STRUCTURE-AWARE ATTRIBUTE ENCODER

On top of the computational cost reduction given by the graph coarsening procedure, we build a structure-aware Variational Autoencoder (sVAE) for node attribute compression. sVAE compresses node embeddings $V \in \mathbb{R}^{n' \times r}$ into a latent space $Z \in \mathbb{R}^{n' \times s}$ with $s \ll r$. In contrast to attribute-only autoencoders, using a graph-specific network architecture allows us to preserve both individual node characteristics and neighborhood dependencies in the latent representations.

The encoder $\text{Enc}_\theta \colon (\mathbf{V}, \mathbf{M}) \mapsto (\mu_{\mathbf{Z}}, \sigma_{\mathbf{Z}}^2)$ and decoder $\text{Dec}_\theta \colon (\mathbf{Z}, \mathbf{M}) \mapsto \hat{\mathbf{V}}$ neural networks learn probabilistic mappings between data and latent space. The tuple $(\mu_{\mathbf{Z}}, \sigma_{\mathbf{Z}}^2)$ gives the mean and variance of the Gaussian latent distribution. Both the encoder and decoder use graph convolutional layers (Hamilton et al., 2017). Doing so allows sVAE to incorporate structure-related information during encoding and decoding. Appendix B.3 provides further architectural details.

In standard variational autoencoder fashion (Kingma & Welling, 2014), the training objective combines reconstruction fidelity with Kullback-Leibler (KL) regularization to ensure that the latent distribution approximates a standard Gaussian prior:

$$\mathcal{L}_{\text{sVAE}} = \mathbb{E}^{\mu_{\mathbf{Z}}, \sigma_{\mathbf{Z}}, \mathbf{V}, \mathbf{M}} \left[ \| \text{Dec}_\theta(\mu_{\mathbf{Z}}, \mathbf{M}) - \mathbf{V} \|_2^2 \right] + \beta \cdot \text{KL} \left[ \mathcal{N}(\mu_{\mathbf{Z}}, \text{diag}(\sigma_{\mathbf{Z}}^2)) \, \| \, \mathcal{N}(0, \mathbf{I}) \right]$$

where the hyperparameter $\beta$ controls regularization strength. sVAE is naturally permutation equivariant, as it merely reduces the dimensionality of input data.

### 3.4 SYMMETRY PROPERTIES

Model behavior should be unaffected by the order of nodes and edges within graphs, such that the model can learn effectively without seeing many permutations of each training graph. MAGiC achieves this by having a permutation invariant model architecture and loss. As such, all permutations of an input graph lead to a unique permutation for the corresponding reconstructed graph, and they have the same loss value. Thus, in Appendix C, we prove that:

**Proposition 2.** *MAGiC's end-to-end architecture is permutation invariant for graphs with the unique distances property (Definition 1).*

For it, we first prove the invariance of our reversible coarsening (Proposition 4) and the equivariance of the mixed-type diffusion (Proposition 5). Following an invariant step with an equivariant one, which matches the input permutation in the output, yields an invariant outcome. We also show that:

**Proposition 3.** *MAGiC's mixed diffusion loss is permutation invariant.*

As our sVAE loss is a per-node aggregation, it is also naturally invariant. Consequently, both optimization functions used in our end-to-end model obey invariance.

## 4 EVALUATION

We evaluate MAGiC on six publicly available datasets: three with rich node attributes, one molecular, and two unattributed. Namely, we measure the performance of synthesizing multi-attributed (Table 1), single-attributed (Table 2), and unattributed (Table 3) graphs. Evaluation setup details are in Appendix D.1. We additionally highlight **qualitative results** in Appendix D.2.

**Baselines**: Since MAGiC is the first work investigating graph generation with multiple heterogeneous attributes, we propose multiple baselines that combine two popular generators for tabular data (i.e., node features), the VAE-based TVAE (Xu et al., 2019) and diffusion-based TabDDPM (Kotelnikov et al., 2023), with a modern diffusion graph synthesizer as DiGress (Vignac et al., 2023). We also include tests with the standalone Mixed Diffusion to assess performance without sVAE and coarsening. We include GruM (Jo et al., 2024) alongside DiGress as baselines for experiments on single-attributed molecular graphs. We add two extra structure-only generators as baselines for unattributed graph tests: EDGE (Chen et al., 2023) and GraphLE (Bergmeister et al., 2024).

**Metrics**: Alongside sampling time, our main results investigate the graph *Structure Quality*, node *Attribute Quality*, and their interaction. We measure structure quality via Maximum Mean Distance (MMD) over four metrics: node degree *(Deg.)*, Laplacian spectrum *(Spec.)*, clustering coefficient

| | Dataset/Method | Structure Quality ↓ | | | | Attribute Quality ↑ | | Tgt. Col. MMD ↓ | Downstr. Util. ↑ | Sample Time ↓ | Effective Nodes |
|---|---|---|---|---|---|---|---|---|---|---|---|
| | | Deg. | Spec. | Clus. | Orb. | Shape | Pair Trend | | | | |
| Twitch | DiGress+TVAE | .344 | .039 | .257 | .124 | .867 | .913 | .281 | .0 | 742 | 160 |
| | DiGress+TabDDPM | .317 | .036 | .240 | .215 | .907 | **.971** | .323 | .0 | 741 | 160 |
| | *Mixed Diffusion* | **.010** | **.009** | **.060** | **.055** | **.945** | .957 | **.002** | .796 | 748 | 160/160 |
| | *MAGiC* | .049 | .038 | .176 | .056 | .866 | .930 | .024 | .685 | **294** | 94.9/160 |
| | *MAGiC (Large)* | .020 | .017 | .177 | .050 | .858 | .940 | .009 | .727 | 784 | 155.0/260 |
| Event | DiGress+TVAE | .307 | .073 | .280 | .491 | **.952** | .793 | .202 | .0 | 741 | 160 |
| | DiGress+TabDDPM | .194 | .194 | .263 | .395 | .835 | .710 | .101 | .580 | 742 | 160 |
| | *Mixed Diffusion* | **.005** | **.007** | .196 | .077 | .821 | **.823** | **.002** | .642 | 748 | 160 |
| | *MAGiC* | .014 | .030 | **.157** | **.036** | .760 | .567 | .004 | .616 | **305** | 94.4/160 |
| | *MAGiC (Large)* | .006 | .020 | .162 | .075 | .768 | .520 | .001 | .599 | 826 | 153.5/260 |
| Ogbn-arxiv | DiGress+TVAE | .042 | .032 | >1 | .413 | **.946** | **.975** | .016 | **.777** | 1272 | 160 |
| | DiGress+TabDDPM | .039 | .032 | .967 | .385 | .500 | .529 | .046 | .469 | 1272 | 160 |
| | *Mixed Diffusion* | **.002** | **.006** | **.116** | **.082** | .874 | .966 | **.002** | .703 | 1280 | 160 |
| | *MAGiC* | .015 | .035 | .183 | .155 | .607 | .752 | .009 | .741 | **479** | 94.2/160 |
| | *MAGiC (Large)* | .010 | .025 | .441 | .071 | .561 | .623 | .002 | .706 | 1277 | 153.7/260 |

Table 1: Main results on multi-attributed graphs in terms of graph quality (for structure, attributes, and their combination), downstream tasks, and sampling time (in seconds).

*(Clus.)*, and four-node orbit counts *(Orb.)*. Node attribute quality treats nodes as tabular rows, comparing column distributions *(Shape)* and pairwise correlations *(Pair Trend)*. To assess structure–attribute interaction, we use Target Column MMD *(Tgt. Col. MMD)* and downstream node classification accuracy *(Downstr. Util.)*. For computational and memory costs, we measure sample time (in seconds) and the number of nodes in diffusion. Finally, for unattributed graphs, we report sampling time and Structure MMD; for molecular data, we check Valid, Unique, and Novel ratios.

**Datasets**: We construct experiments for larger graphs with rich node attributes over three public real-world networks. For smaller, single- and un-attributed settings, we use one molecular and two synthetic datasets from previous works, respectively. The chosen multi-attributed sources each contain a single graph with thousands of nodes representing a social (Twitch (Rozemberczki & Sarkar, 2021) and Event (Carroll et al., 2013)) or citation (OGBN-arxiv (Hu et al., 2020)) network. To create our multi-graph datasets, we sample 200 subgraphs of 160 or 260 nodes from each of the three networks, depending on the experiment. The single-attributed dataset is QM9 (Wu et al., 2017), containing over 100K small molecules with up to 9 nodes each.

## 4.1 Graphs with rich node attributes

Table 1 compares MAGiC with combinations of state-of-the-art graph and tabular generators. We also measure the performance of Mixed Diffusion standalone to ablate the joint effect of coarsening and sVAE. Baselines generate 160-node graphs; for MAGiC only, we also include 260-node results.

The table shows MAGiC and Mixed Diffusion outperforming the baselines on modeling structure–attribute interactions, with MAGiC also being $2.5\times$ faster in sampling. MAGiC consistently outperforms the baselines, with an average of $12.9\times$ for target column MMD and $25.2\%$ for downstream utility. The same pattern extends to structure quality metrics. This result reaffirms that incorporating node attributes into the diffusion model enhances edge connectivity modeling. Furthermore, conflicting information from independent structure and attribute generation can be highly detrimental. As observable in the downstream utility results, baselines sometimes can not predict any test node data correctly.

| | Dataset/Method | Utility ↑ | | |
|---|---|---|---|---|
| | | Valid | Unique | Novel |
| QM9 | DiGress | 98.19 | 96.67 | 25.58 |
| | GruM | **99.69** | **96.90** | 24.15 |
| | *Mixed Diffusion* | 99.46 | 96.82 | **36.10** |

Table 2: Results on molecular graphs (single-label edges and nodes). We report baseline measurements from Jo et al. (2024).

Meanwhile, on node attribute metrics, baselines aided by tabular synthesizers deliver performance comparable to Mixed Diffusion. When strictly measuring the statistical properties of attributes, all nodes are considered independent data points. Thus, the tabular models can more effectively

focus their learning efforts on a narrower problem without relationships between individual samples. Nevertheless, Mixed Diffusion fares best under two out of six total experiments and remains second best in three out of the other four, with small gaps compared to the best baseline. Altogether, Mixed Diffusion is best at accounting for structure and node features interaction, with MAGiC always being a close second. MAGiC's coarsening reduces node counts by $> 40\%$ in all cases, leading, together with sVAE, to $\approx 2.5\times$ faster sampling. We fix the model parameters across all methods and batch sizes to the lowest common denominator across all datasets. However, MAGiC allows higher batch sizes within the same memory budget to further increase its relative efficiency.

**Scalability**: In Table 1, we additionally demonstrate the scalability of MAGiC on large multi-attributed graphs of 260 nodes, i.e., *Magic (Large)*. Comparing the results for 260 nodes against the ones for 160 shows that MAGiC scales well, obtaining similar quality on larger graphs as on the smaller counterparts. Moreover, its run time on the larger graphs is very similar to that of baselines in the case of smaller graphs. The coarsening rate remains on par with previous tests at $\approx 40\%$.

## 4.2 SINGLE-ATTRIBUTED AND UNATTRIBUTED GRAPHS

**Single-attributed graphs**: We run Mixed Diffusion on a molecular dataset to check the method's standalone performance and observe quality in line with the state-of-the-art in Table 2. We do not run MAGiC, as the size of the molecules is not large enough to justify coarsening use. In QM9, Mixed Diffusion achieves performance similar to the state-of-the-art under the ratio of valid and unique molecules out of the valid ones, with scores $> 99\%$ and $> 96\%$ respectively. Notably, MAGiC increases the ratio of novel generated molecules from $25.58\%$ to $36.1\%$. We attribute this to the higher diversity in the overall generation process due to the continuous diffusion component on nodes. Note that the relatively low novelty scores come from QM9 modeling a smaller family of molecules, many of which are already in the training data.

**Unattributed graphs**: Finally, in Table 3, we run MAGiC on unattributed graphs to isolate and assess the impact of the structure coarsening. Despite a performance drop compared to the best baselines, MAGiC can model the inherent statistical properties of different graph types. Unlike for multi-attributed experiments, given the comparison across different graph diffusion architectures, we set the largest possible batch size for each specific instance to time the sampling procedure. Under MMD, GruM is the best performing method, with GraphLE, DiGress, and MAGiC following, while EDGE is last by a wide margin. Time-wise, EDGE is orders of magnitude faster but also has the worst quality output. Aside from it, MAGiC is the fastest method,

| Dataset/Method | | Structure Quality $\downarrow$ | | | Sample |
|---|---|---|---|---|---|
| | Deg. | Clus. | Orb. | Spec. | Time $\downarrow$ |
| **Planar** DiGress | .0007 | .0780 | .0079 | .0098 | 127 |
| EDGE | .0761 | .3229 | .7737 | .0957 | **1** |
| GruM | **.0005** | **.0353** | **.0009** | **.0062** | 156 |
| GraphLE | **.0005** | .0626 | .0017 | .0075 | 40 |
| *MAGiC* | .0054 | .1045 | .1615 | .0304 | 34 |
| **SBM** DiGress | .0018 | **.0485** | **.0415** | **.0045** | 663 |
| EDGE | .0279 | .1113 | .0854 | .0251 | **1** |
| GruM | **.0007** | .0492 | .0448 | .0050 | 717 |
| GraphLE | .0119 | .0517 | .0669 | .0067 | 3231 |
| *MAGiC* | .0256 | .0501 | .0543 | .0077 | 195 |

Table 3: Unattributed graph results. Sampling time is in seconds. For other metrics, we report baseline values from Bergmeister et al. (2024) or, in the case of GruM, Jo et al. (2024).

with a lead that becomes even greater as the average graph size increases from Planar to SBM. Specifically, compared to the second fastest method, GraphLE, MAGiC goes from being $1.15\times$ to $3.67\times$ faster. In the bigger dataset, our method also decreases its disadvantage in terms of MMD, performing better than the aforementioned counterpart in two of the four categories. In summary, MAGiC balances synthesis quality while retaining a significant computational cost reduction.

## 5 CONCLUSION

We present MAGiC, the first diffusion framework for efficiently synthesizing graphs with rich node attributes. It features a mixed-type graph diffusion model for attribute and structure generation, alongside an invertible coarsening and structure-aware attribute encoder for lowering computation and memory requirements in diffusion. During evaluation, MAGiC captures attribute and edge interdependencies $12.9\times$ better and improves performance on downstream tasks by $25.2\%$, while reducing memory utilization and sampling time by $2.5\times$ compared to baselines. By supporting arbitrary node attributes and efficient synthesis, MAGiC addresses the issue of privacy-preserving data-sharing for complex domains like social and document networks.

## ETHICS AND REPRODUCIBILITY STATEMENT

**Ethics**: Our proposed graph generative model has broad applications in modeling human interactions on social media, professional networks, or social contagion situations; it also allows modeling molecular structures, relevant for tasks like drug and material science discovery. As a generative model, our solution can alleviate the need for third parties to tap into confidential or privacy-sensitive data directly when answering questions about it (e.g., examine the spread of a disease among different user groups) or help improve productivity (e.g., propose new drug candidates to investigate).

The manuscript includes input from LLMs for minor rephrasing, grammar, and spelling checks.

**Reproducibility**: To ensure the reproducibility of our research, we include the code for the proposed model, configuration files, and datasets in an anonymized repository.

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

## A  NOTATION SUMMARY

Table 4 summarizes the notation used throughout the manuscript to describe the graph representations at different stages in the MAGiC framework.

| Notation | Description |
|---|---|
| $\mathbf{G}^r = (\mathbf{V} \in \mathbb{R}^{n' \times r}, \mathbf{M} \in \{0,1\}^{n' \times n'})$ | original graph with rich node attributes |
| $\mathbf{G}^s = (\mathbf{Z} \in \mathbb{R}^{n' \times s}, \mathbf{M} \in \{0,1\}^{n' \times n'})$ | graph encoded via sVAE ($s \ll r$) |
| $\mathcal{E}$ | set of edge types (including no edge) |
| $\mathbf{G} = (\mathbf{X} \in \mathbb{R}^{n \times f}, \mathbf{E} \in \mathcal{E}^{n \times n})$ | coarsened graph $n \approx \frac{n'}{2}, f = 2r$ |
| $T$ | total number of diffusion time steps |
| $\mathbf{G}^t = (\mathbf{X}^t \in \mathbb{R}^{n \times f}, \mathbf{E}^t \in [0,1]^{n \times n \times |\mathcal{E}|})$ | graph $G$ with one-hot encoded edges after $t$ noise steps |
| $e_{ij}^t \in [0,1]^{|\mathcal{E}|}$ | edge-type distribution between nodes $i$ and $j$ |
| $\boldsymbol{\epsilon} \sim \mathcal{N}(\mathbf{0}, \mathbf{I})$ | node noise sample |
| $\alpha_t$ | node signal rate at time $t$ |
| $\bar{\alpha}_t$ | node cumulative signal rate up to time $t$ |
| $\mathbf{Q}^t \in [0,1]^{|\mathcal{E}| \times |\mathcal{E}|}$ | edge transition matrix at time $t$ |
| $\bar{\mathbf{Q}}^t \in [0,1]^{|\mathcal{E}| \times |\mathcal{E}|}$ | edge cumulative transition matrix up to time $t$ |
| $q_{\mathbf{G}/\mathbf{X}/\mathbf{E}}(\cdot)$ | forward diffusion process for graphs/nodes/edges |
| $\phi_\theta(\cdot)$ | learnt reverse diffusion process |
| $\mathcal{C}$ | coarsening function |
| $\mathcal{C}$ | coarsening function |
| $x_i$ | node $i$ embedding |
| $\text{Enc}_\theta/\text{Dec}_\theta$ | sVAE encoder/decoder |
| $\pi$ | permutation of a graph |

Table 4: Summary of the main notation used in the main text.

## B  MODEL DETAILS

Below, we provide details on the coarsening algorithm and learning setup of MAGiC.

### B.1  IMPLEMENTATION NOTES

The following describes our procedure for training a model that harnesses mixed-type diffusion as a backbone, alongside coarsening and structure-aware node attribute encoding via sVAE. Before training the diffusion model, we pretrain the sVAE used to reduce attribute dimensionality, then apply the sVAE, followed by the coarsening to the training data in preparation. Thus, the diffusion

---

**Algorithm 3** Graph Coarsening

    **Input**: $G = (\mathbf{Z} \in \mathbb{R}^{n \times s}, \mathbf{M} \in \{0,1\}^{n' \times n'})$
1: $\mathbf{X} \leftarrow \varnothing, \mathcal{M} \leftarrow \varnothing$
2: **for** $m_{ij} \leftarrow$ edges in $\mathbf{M}$ sorted by $\mathcal{D}(z_i, z_j)$ **do**
3:    **if** $\mathbf{z}_i \in \mathbf{Z} \wedge \mathbf{z}_j \in \mathbf{Z}$ **then**
4:       $\mathbf{X} \leftarrow \mathbf{X} \cup \text{Concat}(\mathbf{z}_i, \mathbf{z}_j)$
5:       $\mathbf{Z} \leftarrow \mathbf{Z} \setminus \{\mathbf{z}_i, \mathbf{z}_j\}$
6:    **else**
7:       $\mathcal{M} \leftarrow \mathcal{M} \cup m_{ij}$
8:    **end if**
9: **end for**
10: **for** $\mathbf{z}_i \in \mathbf{Z}$ sorted by $\mathcal{D}(z_i, \mathbf{0})$ **do**
11:    $\mathbf{X} \leftarrow \mathbf{X} \cup \text{Concat}(\mathbf{z}_i, \mathbf{0})$
12: **end for**
13: $\mathbf{E} \leftarrow \mathbf{0}^{|\mathbf{X}| \times |\mathbf{X}|}$
14: **for** $m_{ij} \leftarrow \mathcal{M}$ **do**
15:    $a \leftarrow \text{ParentID}(\mathbf{z}_i, \mathbf{X}), \; b \leftarrow \text{ParentID}(\mathbf{z}_j, \mathbf{X})$
16:    $\mathbf{E}_{ab} \leftarrow \mathbf{E}_{ab} + \text{EncodeEdge}(m_{ij})$             { Figure 3b}
17: **end for**
18: **return** $\mathbf{X}, \mathbf{E}$

---

loss is optimized directly in the reduced embedding space, and the mapping to the original space only happens when a complete output is required, like during evaluation. In doing so, we avoid involving the inverse coarsening and sVAE decoder while training the mixed-type diffusion model so as not to increase training costs.

### B.2 COARSENING ALGORITHM

Algorithm 3 describes the coarsening steps. We begin by sorting edges based on the distance between their constituent nodes in non-decreasing order (line 2). Based on the resulting order, we greedily merge connected node pairs (lines 3–8). We then create zero-padded pairs with all nodes that have not been merged (lines 10–11). Finally, with all the pairs in place, we create edges between all newly created nodes according to Figure 3b (lines 14–16).

Algorithm 4 provides more details on the decoarsening. We first split each aggregated node representation into two nodes (line 3), adding them and their edge to the original graph while skipping any dummy zero-filled nodes (lines 4–8). Subsequently, we decode the edge type to expand each edge in the compressed graph to the original graph edges it aggregates, adding them to the new graph structure (lines 10–14).

### B.3 sVAE

Figure 4 visualizes the architecture of sVAE for the case of two encoding and decoding layers, respectively. Each layer takes as input a representation of the node attributes after the previous step, along with the connectivity information of the graph. As is typical in VAEs, the encoder $Enc_\theta$ estimates the parameters of a prior distribution, which, in our case, are the mean and variance of a Gaussian. Consequently, the decoder $Dec_\theta$ expects a sample drawn from a Gaussian latent distribution as input. Furthermore, we change between a sigmoid or softmax activation function for each attribute based on whether it represents a numerical value or part of a one-hot encoded categorical. If node attributes do not originally encode a tabular data row, we consider each feature a unique numerical column.

## C PROOFS

For an input graph $\mathbf{G} = (\mathbf{X} \in \mathbb{R}^{n \times f}, \mathbf{M} \in \mathbb{E}^{n \times n})$ with $n$ nodes, let the bijection $\pi : \{0, 1, \ldots, n - 1\} \rightarrow \{0, 1, \ldots, n - 1\}$ be a permutation of $G$'s nodes. As such, $\pi^{-1}$ is the inverse permutation

---

**Algorithm 4** Inverse Graph Coarsening

**Input**: latent graph $G = (\mathbf{X} \in \mathbb{R}^{n \times f}, \mathbf{E} \in \mathbb{N}_{15}^{n \times n})$
1: $\mathbf{M} \leftarrow \mathbf{0}$, $\mathbf{Z} \leftarrow \varnothing$
2: **for** $\mathbf{x}_k$ in $\mathbf{X}$ **do**
3: $\quad (\mathbf{z}_i, \mathbf{z}_j) \leftarrow \mathbf{x}_k$
4: $\quad \mathbf{Z} \leftarrow \mathbf{Z} \cup \{\mathbf{z}_i\}$
5: $\quad$ **if** $\mathbf{z}_j \neq \mathbf{0}$ **then**
6: $\quad\quad \mathbf{Z} \leftarrow \mathbf{Z} \cup \{\mathbf{z}_j\}$
7: $\quad\quad M_{ij} = 1$
8: $\quad$ **end if**
9: **end for**
10: **for** $\mathbf{e}_{ij} \leftarrow \mathbf{E}$ **do**
11: $\quad$ **for** $(a, b) \in \text{DecodeEdge}(\mathbf{e}_{i,j})$ **do**
12: $\quad\quad M_{ab} = 1$
13: $\quad$ **end for**
14: **end for**
15: **return** $\mathbf{Z}, \mathbf{M}$

---

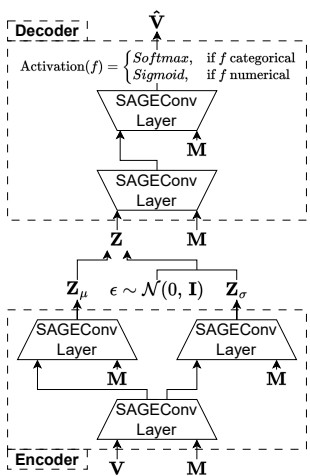

Figure 4: sVAE architecture with two encoder and decoder layers each.

obeying $\pi^{-1}(\pi(i)) = i \,\forall\, i \in \{0, 1, \dots, n-1\}$. Applying $\pi$ to $G$ gives $\pi.G = (\pi.\mathbf{X}, \pi.\mathbf{E})$ where $(\pi.\mathbf{X})_i = \mathbf{X}_{\pi^{-1}(i)}$ and $(\pi.\mathbf{M})_{ij} = \mathbf{M}_{\pi^{-1}(i)\pi^{-1}(j)} \,\forall\, i, j \in \{0, 1, \dots, n-1\}$.

Given the distance metric $\mathcal{D}(x_i, x_j)$ from the main text, we define:

**Definition 1** (Unique distances property)**.** *The unique distances property denotes any graph in which all neighboring node pairs $(x_i, x_j)$ have:*

*(i) unique distances as defined by $\mathcal{D}(x_i, x_j)$, allowing unique inter-pair ordering;*
*(ii) $\mathcal{D}(x_i, \mathbf{0}) \neq \mathcal{D}(x_j, \mathbf{0})$, allowing unique intra-pair ordering.*

A function mapping between two graphs is permutation equivariant if its output permutes in accordance with its input. Contrastingly, it is permutation invariant if its output remains unchanged regardless of how the input node ordering.

**Proposition 4.** *The composition of MAGiC's sVAE encoder and coarsening is permutation invariant for graphs with the unique distances property.*

*Proof Proposition 4.* The composition of MAGiC's sVAE encoder and coarsening is permutation invariant if both sampling from the sVAE encoder Enc$_\theta$ output and coarsening $\mathcal{C}$ are invariant or one is invariant while the other one is equivariant.

The structure-aware encoder forming $\mathrm{Enc}_\theta$ updates only the representations of nodes, mirroring any permutation of the input nodes in the output. Also, sampling from the latent distribution only changes the node embedding content, not node order. Thus, sVAE (and its sampling) is equivariant.

The coarsening of $\mathcal{C}$ creates nodes in the output graph by iterating through neighboring input node pairs from lowest to highest based on $\mathcal{D}(\cdot,\ \cdot)$. Since all distances between neighbors are assumed to be different (Definition 1 (i)), the ordering of the pairs is fixed, regardless of the original ordering. Within each pair, the chosen order for merging nodes affects the value of the merged node's embedding and its local connectivity. Since nodes within a pair are ordered by $\mathcal{D}(\cdot,\ \mathbf{0})$, and are assumed always to have different values (Definition 1 (ii)), the ordering of nodes within a pair is also fixed, regardless of the original ordering. Thus, coarsening is invariant.

From the above, it follows that, under the specified assumptions on $\mathrm{Diff}$ and $\mathrm{Mag}$, the composition MAGiC's sVAE encoder and coarsening.

$\square$

**Proposition 5.** *MAGiC's mixed-type diffusion is permutation equivariant.*

*Proof Proposition 5.* Starting from the input, the diffusion model architecture involves, in sequence: augmenting the input graph representation with additional structural features, applying a node/edge-wise multi-layer perceptron block, applying one or more graph transformer blocks, and applying a final node/edge-wise multi-layer perceptron block. Similar to Vignac et al. (2023), per-node structural features are permutation equivariant. Likewise, the per-graph structural features are permutation invariant (trivially, since the graph itself is the only element in the sequence). The multi-layer perceptron block is permutation equivariant, and so are the individual transformer layers with their self-attention mechanism.

Thus, the mixed-type diffusion of MAGiC is permutation equivariant.

$\square$

*Proof Proposition 1.* Let $\mathbf{Z}_a \prec \mathbf{X}_i$ denote that the node embedding $\mathbf{Z}_a$ is present in $\mathbf{X}_i$. The inverse coarsening is a function $\mathcal{C}^{-1} = (\mathbf{X} \in \mathbb{R}^{n \times f},\ \mathbf{E} \in \mathcal{E}^{n \times n}) \to \mathbb{R}^{n' \times s},\ \mathbf{M} \in \{0,1\}^{n' \times n'})$. Let $\mathbf{G}^s = (\mathbf{Z},\ \mathbf{M})$ and $\mathbf{G} = (\mathbf{X},\ \mathbf{E})$ be two undirected graphs such that $\mathcal{C}(\mathbf{G}^s) = \mathbf{G}$. Specifically, in terms of node features, $\mathcal{C}$ transforms:

$$\mathbf{Z} = \begin{pmatrix} \mathbf{Z}_1 \\ \mathbf{Z}_2 \\ \vdots \\ \mathbf{Z}_{n'} \end{pmatrix} \quad \& \quad \mathbf{M} = \begin{pmatrix} \mathbf{M}_{1\ 1} & \mathbf{M}_{1\ 1} & \dots & \mathbf{M}_{1\ n'} \\ \mathbf{M}_{1\ 2} & \mathbf{M}_{2\ 2} & \dots & \mathbf{M}_{2\ n'} \\ \vdots & \vdots & \vdots & \vdots \\ \mathbf{M}_{1\ n'} & \mathbf{M}_{2\ n'} & \dots & \mathbf{M}_{n\ n'} \end{pmatrix}$$

into:

$$\mathbf{X} = (\mathbf{X}_1, \mathbf{X}_2, \dots, \mathbf{X}_p, \mathbf{X}_{p+1}, \mathbf{X}_{p+2}, \dots, \mathbf{X}_{p+s})^{\mathrm{tr}} \text{ where}$$
$$p + s = n'$$
$$\mathbf{X}_{i \leq p} = \min_{\mathcal{D}} \{\mathrm{Concat}(\mathbf{Z}_a,\ \mathbf{Z}_b) \mid a < b;\ \mathbf{M}_{a\ b} = 1;\ \mathbf{Z}_a, \mathbf{Z}_b \notin \mathbf{X}_{j \leq i};\ \mathcal{D}(\mathbf{Z}_a,\ \mathbf{0}) < \mathcal{D}(\mathbf{Z}_b,\ \mathbf{0})\}$$
$$\mathbf{X}_{p < i \leq n'} = \mathrm{Concat}(\min_{\mathcal{D}} \{\mathbf{Z}_a \mid \mathbf{Z}_a \notin \mathbf{X}_{j \leq i}\},\ \mathbf{0})$$
$$\forall a,\ \exists!\ i : \mathbf{Z}_a \prec \mathbf{X}_i$$

and:

$$\mathbf{E} = \begin{pmatrix} \mathbf{E}_{1\ 1} & \mathbf{E}_{1\ 1} & \dots & \mathbf{E}_{1\ n} \\ \mathbf{E}_{1\ 2} & \mathbf{E}_{2\ 2} & \dots & \mathbf{E}_{2\ n} \\ \vdots & \vdots & \vdots & \vdots \\ \mathbf{E}_{1\ n} & \mathbf{E}_{2\ n} & \dots & \mathbf{E}_{n\ n} \end{pmatrix} \text{ where}$$

$$\mathbf{E}_{ij} = 2^0 \cdot \mathbf{M}_{ac} + 2^1 \cdot \mathbf{M}_{ad} + 2^2 \cdot \mathbf{M}_{bc} + 2^3 \cdot \mathbf{M}_{bd}, i < j,\ \forall\ (\mathbf{X}_i = (\mathbf{Z}_a,\ \mathbf{Z}_b),\ \mathbf{X}_j = (\mathbf{Z}_c,\ \mathbf{Z}_d))$$

Note: coarsening creates as many pairs $p$ as possible, leaving a minimum number of single nodes $s$.

Reversing the function starts with finding all the node representations in the coarse input of the form $\mathbf{X}_s = \mathrm{Concat}(\mathbf{Z}_a,\ \mathbf{0}) \in \mathbf{X}$, where the $\mathbf{Z}_a$ entries are the single nodes of $\mathbf{Z}$. The remaining

node representations $\mathbf{X}_s = \text{Concat}(\mathbf{Z}_a, \mathbf{Z}_b) \in \mathbf{X}$, contain in $\mathbf{Z}_a$ and $\mathbf{Z}_b$ the representations of all other nodes from $\mathbf{Z}$ which have been merged. Consequently, recovering edges starts with adding an edge $\mathbf{M}_{ab} = 1$ between all nodes denoting a pair $\mathbf{X}_s = Concat(\mathbf{Z}_a, \mathbf{Z}_b) \in \mathbf{X}$. Furthermore, from adjacency entries for edges $\mathbf{E}_{ij} \neq 0$, $i < j$ with $\mathbf{X}_i = \text{Concat}(\mathbf{Z}_a, \mathbf{Z}_b)$, and $\mathbf{X}_j = \text{Concat}(\mathbf{Z}_c, \mathbf{Z}_d)$, all bidirectional edges between $\mathbf{Z}_a$ / $\mathbf{Z}_b$ and $\mathbf{Z}_c$ / $\mathbf{Z}_d$ can be recovered from the corresponding active bits in the binary encoding of $\mathbf{E}_{ij}$.

Thus, a permutation $\pi$ of the original nodes $\mathbf{V}$ and edges $\mathbf{M}$ is recovered, meaning that MAGiC's coarsening is invertible up to permutation.

$\square$

*Proof Proposition 2.* The end-to-end architecture is a composition of the sVAE and coarsening, alongside the mixed-type diffusion. The first two jointly are invariant by Proposition 4, while the mixed-type diffusion is equivariant by Proposition 5.

The invariant sVAE and coarsening combination maps all input permutations to a single output ordering, while the equivariant mixed-type diffusion retains that ordering when applied. As such, all permutations of the original graph get mapped to a single permutation after the sVAE, coarsening, and diffusion pipeline, making the end-to-end MAGiC invariant.

$\square$

*Proof Proposition 3.* The loss' invariance can be proven by computation:

$$
\begin{aligned}
L((\pi.\hat{\epsilon};\ \pi.\hat{p}_E),\ (\pi.\epsilon;\ \pi.\mathbf{E})) &= ||\pi.\hat{\epsilon} - \pi.\epsilon||^2 + \lambda CrossEntropy(\pi.\hat{p}_E, \pi.\mathbf{E}) && \text{def.} \\
&= ||\hat{\epsilon} - \epsilon||^2 + \lambda CrossEntropy(\pi.\hat{p}_E, \pi.\mathbf{E}) && \textit{MSE invar.} \\
&= ||\hat{\epsilon} - \epsilon||^2 + \lambda CrossEntropy(\hat{p}_E, \mathbf{E}) && \textit{CrossEntropy invar.} \\
&= L((\hat{\epsilon};\ \hat{p}_E),\ (\epsilon;\ \mathbf{E})) && \text{def.}
\end{aligned}
$$

$\square$

# D    EVALUATION EXTRAS

Below we provide additional details related to evaluation.

## D.1    SETUP INFORMATION

We run all our experiments on an Nvidia RTX 4090 GPU with 24 GB of memory. For the Twitch and Event datasets, our target columns indicate whether a user may earn money from the platform and whether the gender of a user is marked as female. In OGBN-arxiv, initial node attributes are 128-dimensional embeddings of a scientific article title and abstract. Within the evaluation metrics for node attributes, we interpret each embedding entry as a numerical column in a tabular data row. For OGBN-arxiv, the created label denotes whether a paper is registered to one of the top four most popular categories. Based on preliminary tests, we set MAGiC's sVAE compression factor $f' = \lfloor \frac{f}{4} \rfloor$ in all experiments for a good trade-off between compression and quality. For the above datasets with rich node attributes, we train the diffusion model for each experiment over 5000 epochs and weight the cross-entropy of edges 5 times higher than the MSE for node attributes in MAGiC and Mixed Diffusion across all experiments. The training of sVAE happens before that of the diffusion model for a maximum of 5000 epochs on the same training dataset, with an early stopping applied based on the validation set loss. On the QM9 molecular datasets, we train our method for 1k epochs. As for unattributed graphs, we run our method for 100k epochs in planar and 28k in SBM. Further hyperparameters and experimental configuration settings are present in our codebase.

## D.2    QUALITATIVE RESULTS

Table 5 shows real sample graphs from the Twitch and Event datasets. Table 6 and Table 7 showcase an example graph from the two datasets for our baselines and proposed methods, respectively. We only show the node feature values of the first 10 nodes for readability. Node colors represent the relative connectivity density of nodes (blue = lowest, red = highest).

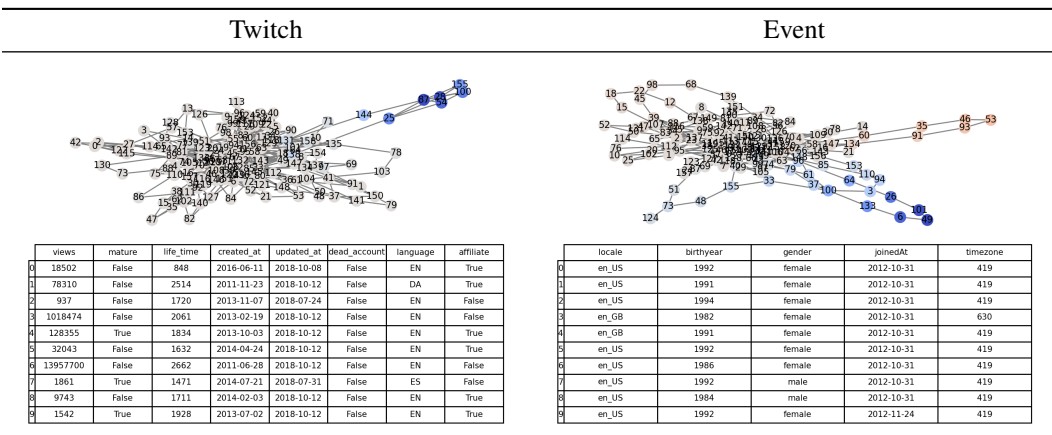

Table 5: Real samples from the Twitch and Event datasets.

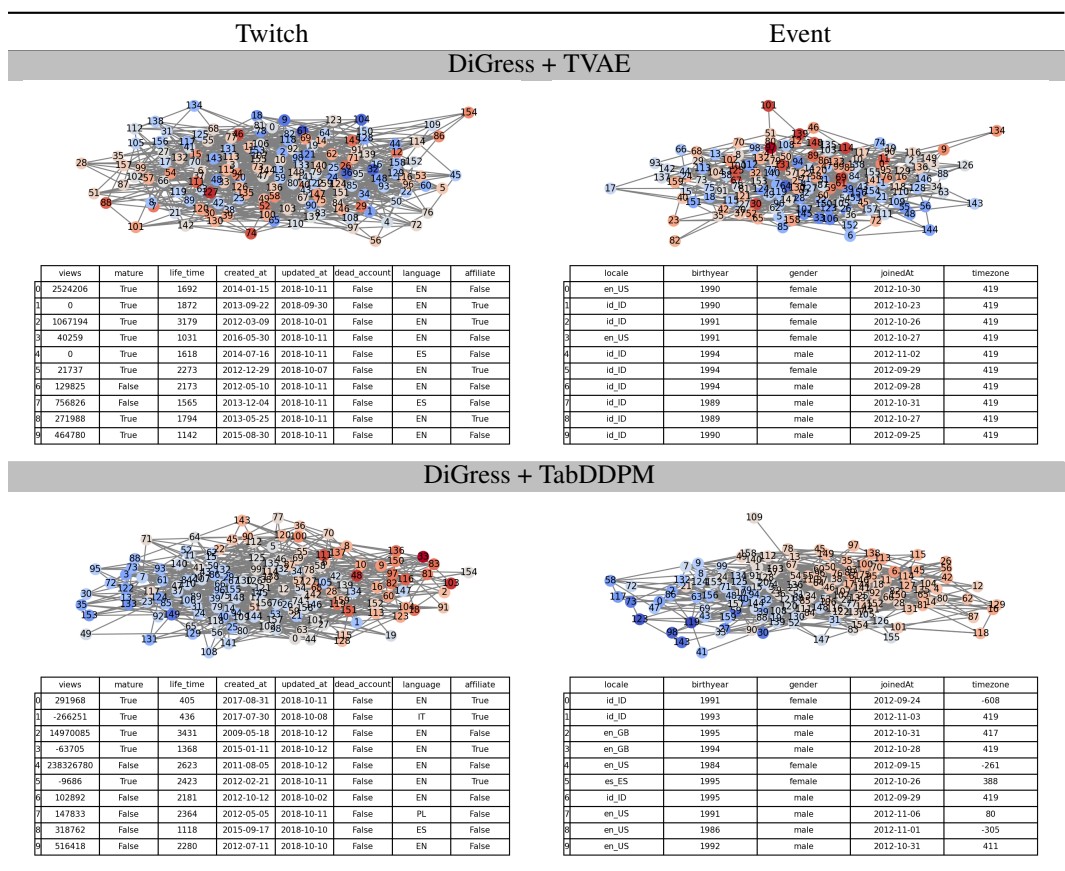

Table 6: Samples from the Twitch and Event datasets generated by the two baselines: DiGress + TVAE and DiGress + TabDDPM.

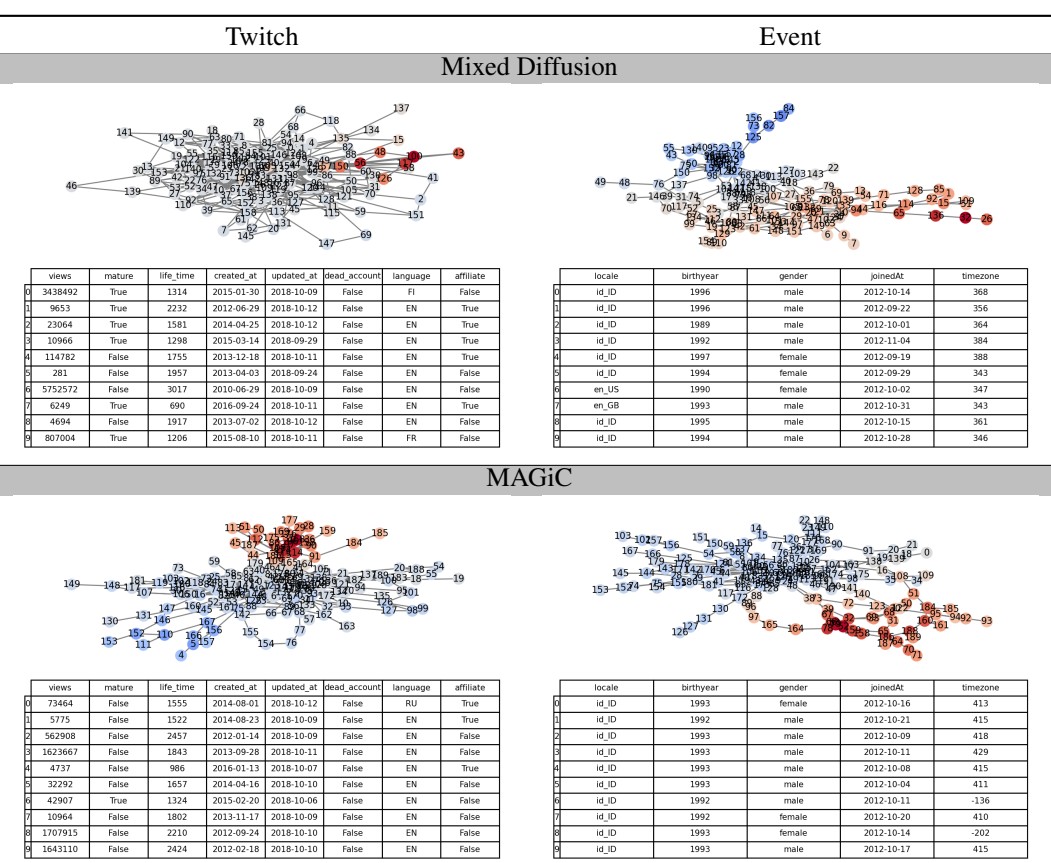

Table 7: Samples from the Twitch and Event datasets generated by our two proposed methods: Mixed Diffusion and the full MAGiC.

