# OpenReview forum: "MAGiC: Attributed Graph Generation via Mixed-type Diffusion and Coarsening"
_ICLR.cc/2026/Conference — ICLR 2026 Conference Withdrawn Submission_

### Official Review · Reviewer_psLA · 2025-10-22

**Soundness:** 1
**Presentation:** 1
**Contribution:** 1
**Rating:** 2
**Confidence:** 4

**Summary:**

This paper introduces a generative model for *mixed-type* graph diffusion, addressing graphs with continuous node attributes and discrete edge attributes. The work proposes two additional methodological contributions: an invertible graph coarsening mechanism and a structure-aware variational autoencoder (VAE). A key theoretical assumption underlying the approach is that graphs in the dataset satisfy a *unique distance* property, meaning that all pairwise node distances, as well as distances between nodes and a designated origin, are distinct.

**Strengths:**

The paper tackles an underexplored yet important problem setting. While most prior works on graph diffusion focus on discrete node and edge attributes, this work extends the paradigm to continuous node attributes, a valuable and challenging scenario for real-world applications.

The paper also introduces a new task formulation together with dedicated datasets and experiments, thereby contributing useful resources and baselines to the community. These efforts have the potential to stimulate further research in mixed-type graph diffusion.

**Weaknesses:**

**Novelty and Originality**
The overall methodological novelty appears limited. The proposed “mixed diffusion” backbone essentially combines existing discrete and continuous diffusion mechanisms without deeply addressing their interaction. In particular, the model does not explicitly account for the fact that discrete diffusion induce abrupt structural changes in the graph, whereas continuous diffusion evolves smoothly. Clarifying how the model reconciles these fundamentally different dynamics would strengthen the contribution.

**Unique Distance Assumption**
A central theoretical assumption of the paper is the *unique distance* property. However, this assumption is only explicitly stated in Proposition 2, despite being required for Propositions 1 and 3 as well. Moreover, its formal definition appears only in the Appendix. Given that this assumption underpins much of the paper’s theoretical reasoning, it should be clearly introduced and discussed in the main text.

Additionally, several datasets used in the experiments, such as unttributed graphs or those with discrete node attributes (e.g., QM9), do not satisfy this assumption. The paper does not explain how such cases are handled, which undermines the practical validity of the results. Clarifying how the method is applied when the unique distance property does not hold would considerably improve the paper’s coherence and transparency.

**Experimental Evaluation**
The experimental section does not fully align with the paper’s stated scope. Although the method is designed for graphs with continuous node attributes, half of the datasets used (3 out of 6) contain no or only discrete node attributes.

In the continuous-node setting, where no established baselines exist, the proposed MAGIC model is outperformed by the mixed diffusion baseline in terms of sample quality, although MAGIC achieves faster sampling. In the remaining benchmarks, the model is consistently outperformed by GRUM, which itself is not state-of-the-art compared to more recent methods such as DeFog, CatFlow, or SID.

Furthermore, novelty evaluation on QM9 is problematic, as QM9 represents an exhaustive enumeration of small organic molecules. It would be helpful if the authors could justify or reconsider the use of novelty as a metric in this context.

**Clarity and Presentation**
The paper’s presentation is at times difficult to follow. Several core definitions and claims are relegated to the Appendix, which hinders readability.

The coarsening algorithm is introduced, but its role in the full sampling process remains unclear—particularly whether sampling is performed from the coarse graph or directly through mixed diffusion. A more explicit description of the complete generation pipeline would be beneficial.

Finally, the use of a VAE on node embeddings presupposes independence among node representations, which is not a trivial assumption for graph-structured data. Providing theoretical or empirical justification for this assumption would strengthen the paper’s validity.

-----

**DeFog**: Yiming Qin, Manuel Madeira, Dorina Thanou, and Pascal Frossard. Defog: Discrete flow matching
for graph generation. In Forty-second International Conference on Machine Learning, 2025. URL
https://openreview.net/forum?id=KPRIwWhqAZ.

**CatFlow**: Floor Eijkelboom, Grigory Bartosh, Christian A. Naesseth, Max Welling, and Jan-Willem van de
Meent. Variational flow matching for graph generation. In The Thirty-eighth Annual Confer-
ence on Neural Information Processing Systems, 2024. URL https://openreview.net/
forum?id=UahrHR5HQh.

**SID**: Yoann Boget. Simple and critical iterative denoising: A recasting of discrete diffusion in graph gen-
eration. In Proceedings of the 42th International Conference on Machine Learning, Proceedings
of Machine Learning Research. PMLR, July 2025.

**Questions:**

1. **Regarding the theoretical assumptions:**
Could the authors clarify whether Definition 1 (the *unique distance* assumption) is indeed required for Propositions 1 and 3, in addition to Proposition 2? If so, it would be helpful to make this dependency explicit in the main text.

2. **On the coarsening algorithm:**
Could the authors elaborate on the role of the proposed graph coarsening algorithm within the overall sampling procedure? In particular, how is the coarse graph generated?

3. **On the validity of the unique distance assumption:**
How does the proposed model handle graphs that do not satisfy the *unique distance* property? A clarification on whether any relaxation, approximation, or preprocessing is applied in such cases would be greatly appreciated.

---

### Official Review · Reviewer_nLE5 · 2025-11-03

**Soundness:** 3
**Presentation:** 2
**Contribution:** 2
**Rating:** 2
**Confidence:** 4

**Summary:**

This paper proposes MAGiC, a mixed-type diffusion framework for attributed graph generation. In detail, the method applies discrete diffusion for graph structure generation and continuous diffusion for node attribute generation. The model aims to improve the scalability of diffusion-based graph generation while handling graphs with rich continuous attributes.

**Strengths:**

- The paper presents an interesting integration of discrete and continuous diffusion processes within a unified framework, enabling graph generation with mixed-type node attributes.
- The paper includes theoretical proofs of permutation invariance and invertibility, which enhance its methodological soundness.

**Weaknesses:**

- **Inferior graph generation performance**: In Table 1, the Mixed Diffusion sometimes outperforms MAGiC, and in Table 3, other diffusion-based methods such as DiGress and GruM achieve superior results. Moreover, Table 2 lacks the reported performance of MAGiC itself, showing only baselines. Efficiency alone cannot be considered a sufficient contribution—especially since MAGiC’s sampling time would likely remain slower than that of autoregressive models such as GRAN [1] or GEEL [2]. I am aware that GRAN and GEEL are not for graph generation with continuous attributes but the authors can implement this as they have done for DiGress.
- **Lack of clarity in figures**: Several figures are difficult to interpret. For example, the meaning of $M_i$ in Figure 3 is not explained, and Figure 2 lacks a clear legend indicating what each color represents. More detailed captions and annotations are needed to make the visualizations self-contained.
- **Lack of explanation on experimental metrics**: The attribute quality metric is non-standard in graph generation literature, but the paper provides only a brief description. Additional explanation is needed regarding what this metric measures and how it is computed. Similarly, the target column metric and sampling time results require clarification (e.g., which computational resources were used, and how many graphs were generated).

[1] Liao, R., et al. Efficient graph generation with graph recurrent attention networks. NeurIPS 2019.

[2] Jang, Y., et al. A simple and scalable representation for graph generation. ICLR 2024.

**Questions:**

- How exactly does MAGiC achieve improved scalability compared to previous diffusion-based graph generators? Is this solely due to the coarsening, or are there additional architectural optimizations?
- Why does the method focus exclusively on node attributes, without addressing edge attributes? In molecular generation, for instance, bond types (single, double, aromatic) are critical. How are these handled in *Table 2* experiments?
- What is the practical application of graph generation with continuous node labels (e.g., for social networks)? The current examples feel somewhat artificial. Potentially, molecular conformation generation tasks could serve as more meaningful use cases—why are such experiments not included?

---

### Official Review · Reviewer_XEQj · 2025-11-03

**Soundness:** 3
**Presentation:** 3
**Contribution:** 3
**Rating:** 6
**Confidence:** 4

**Summary:**

The authors tackle the problem of synthetic graph generation in the challenging setting of heterogenous graphs with large feature nodes and graph size. In particular, a novel mixed-type diffusion is proposed for addressing both discrete and continuous features and applied to 6 real-world datasets to showcase the benefits of the proposed approach.

**Strengths:**

* A novel architecture for the diffusion model is proposed which combines continous noise over node embeddings and discrete noise over graph structure and a joint denoising process. In addition, a novel invertible coarsening is proposed to speed up the diffusion process enabling handling of larger graphs. A structural encoder for the node attributes utilizing the neighbor information.
* Invariance of the graph input is shown for the model architecture of MAGIC is shown in the paper and includes the components of coarsening and mixed-type diffusion.
* Experiments on a wide-variety of datasets including rich node attributes, and a couple unattributed ones. Ablation studies are also conduction to show the effects of the mixed diffusion and also including coarsening/structural encoding. In most cases, a healthy balance between scalability, computation time, and quality of generated graphs is shown compared to the state-of-the-art approaches.

**Weaknesses:**

(a) It is unclear how to balance the trade-offs between the structural quality versus the generation time.
(b) The improvements in the size of the graphs that MAGIC can handle seems to be roughly 2x the previous methods which seems a bit modest.

**Questions:**

(1) Is it the case that  mixed diffusion is the primary component which provides the quality improvement and the other components of structure encoding/coarsening providing the speedup and handling of larger graphs based on the ablation studies?

(2) Can the graph coarsening approach be continued further and combine more number of  nodes/edges? Would we able to handle much larger graphs and trade-off quality of the generated graphs?

---

### Official Review · Reviewer_845E · 2025-11-05

**Soundness:** 2
**Presentation:** 2
**Contribution:** 2
**Rating:** 2
**Confidence:** 4

**Summary:**

The authors propose a method called MAGiC (Mixed-type Attributed Graph Diffusion model with Coarsening), a framework that enables graph diffusion with mixed-type diffusion — combining discrete diffusion for graph structure and continuous diffusion for node attribute embeddings within a single model. It also introduces an invertible coarsening algorithm and a structure-aware attribute encoder.

**Strengths:**

The idea of combining discrete and continuous diffusion models is important and timely. However, the research analysis, insights, and overall presentation could be improved. More comments are provided below.

**Weaknesses:**

* The paper is difficult to follow and requires multiple readings to understand the main ideas.

* Several terms such as lossless reduction and effective nodes are not clearly explained, leaving the reader to infer their meanings.

* The overall results for MAGiC appear relatively weak compared to Mixed Diffusion, suggesting that the model captures only limited aspects of the data. For instance, when generating a graph with around 260 nodes, the model captures approximately 155 nodes. This section is unclear and seems to indicate that MAGiC struggles to accurately reconstruct coarse structures into large graphs, thereby limiting the effectiveness of its contribution.

* Table 3 shows poor performance of MAGiC on unattributed graphs, indicating that it may only perform well on attributed graphs and thus lacks generalizability to other graph types. This limits the model’s ability to handle diverse graph structures.

* There are inaccurate claims such as “MAGiC being the fastest method” (lines 470–471). Table 3 clearly shows that its sampling time (in seconds) is at least 30× slower than EDGE. The paper also fails to explain why EDGE is faster than MAGiC or which components in the MAGiC contribute to its slower performance.

* The authors claim that the reconstruction algorithm after coarsening is lossless (lines 69–70). Does this imply that each graph has a unique one-to-one mapping before and after coarsening? If so, it would be valuable to include an analysis demonstrating this property. Such an evaluation is currently missing and would significantly benefit the community by clarifying the theoretical guarantee of lossless reconstruction.

**Questions:**

see above.

---

### Note · Authors · 2025-11-12

I have read and agree with the venue's withdrawal policy on behalf of myself and my co-authors.